# High-Intensity Interval Exercise Drives Vitamin D Receptor Expression in Skeletal Muscle via Recruitment of Non-Parenchymal Cells, Not Upregulation in Muscle Fibers

**DOI:** 10.3390/nu17233733

**Published:** 2025-11-28

**Authors:** Kenneth Ladd Seldeen, Ni Wang, Rupadevi Muthaiah, Owen Paul Treanor, Anna Leigh Davis, Lee Daniel Chaves, Ramkumar Thiyagarajan, Brandon J. Marzullo, Donald Albert Yergeau, Bruce Robert Troen

**Affiliations:** 1Division of Geriatrics & Landon Center on Aging, University of Kansas Medical Center, Kansas City, KS 66103, USA; 2Research Service, Veterans Affairs Kansas City Healthcare System, Kansas City, MO 64128, USA; 3New York State Center of Excellence Bioinformatics & Life Sciences, University at Buffalo, Buffalo, NY 14068, USA

**Keywords:** aging, exercise, satellite cells, macrophages, vitamin D receptor

## Abstract

**Background:** High-intensity interval exercise (HIIE) is gaining interest as an alternative to traditional moderate-intensity exercise due to its shorter exercise regimens. Yet, it still induces significant muscular adaptations, including metabolic remodeling, enhanced mitochondrial biogenesis, and improved endurance capacity. Exercise has been shown to increase vitamin D receptor (VDR) expression acutely; however, the role of this effect and whether it occurs during HIIE remain to be elucidated. **Objectives/Methods**: Here, we investigated the time-dependent effects of a single bout of high-intensity interval exercise (HIIE) on systemic inflammatory cytokine profiles and gene expression, including VDR, in aged skeletal muscle. Sedentary aged mice (male C57Bl/6J at 24 months of age) were provided a 10-min HIIE session, and blood and tissues were harvested at 1-, 4-, and 24-h post-exercise, and compared with sedentary mice. **Results**: Our findings indicate that HIIE elicits a transient systemic inflammatory response peaking at 4 h post-exercise and returning to pre-exercise levels by 24 h. Using principal component analysis, we identified a similar pattern in the mRNA profiles, with clear clusters separating sedentary groups at 1 and 4 h after acute HIIE, but not after 24 h. Although VDR mRNA follows this pattern, protein expression, as determined by Western blot and immunohistochemical analysis, reveals persistence at 24 h. As VDR was localized to the periphery of muscle fibers, we investigated and found that VDR co-localizes with PAX7 (a marker for satellite cells) and F4/80-expressing macrophages. This suggests that the observed increase in VDR expression following exercise may be attributed to ancillary cell response during muscle remodeling. **Conclusions**: Together, these results provide novel insights into the transient molecular changes occurring 1 and 4 h after HIIE, which subsequently return to baseline after 24 h. This highlights the potential of HIIE in muscle adaptation and recovery, particularly in older individuals.

## 1. Introduction

Exercise is an important intervention for frailty [1,2,3,4,5], a multifactorial clinical syndrome that increases risk for morbidity and mortality [6]. Despite the well-established benefits of exercise, only 8% of individuals aged 75 and older meet the guidelines of 30 min of daily exercise plus strength training—a concerning statistic, as regular exercise is vital for the health of older adults. Major barriers to physical activity include a lack of time and the lengthy duration of exercise sessions [7,8]. As an alternative, high-intensity interval exercise (HIIE), characterized by repeated bouts of intense exercise interspersed with lower-intensity active recovery [9], is a promising exercise option that elicits functional benefits with lower session times [10,11]. Nonagenarians have been demonstrated to respond well to HIIE. Research by Fiatarone et al. (1990) demonstrated that they tolerated HIIE effectively, with notable improvements in strength and gait speed over 10 weeks [12]. Subsequent studies have confirmed the safety and effectiveness of HIIE for individuals over 65, demonstrating improvements in fitness levels, increased insulin sensitivity, enhanced cardiovascular function, and an improved quality of life [13,14,15,16]. Losa-Reyna et al. (2019) found that HIIE significantly reduced frailty in individuals aged 75 and older, highlighting its role in promoting healthier aging [17]. HIIE can also yield benefits with less time, as our research group previously reported that a regimen of short-session HIIE (~10 min) increased functional capacity and reduced frailty in both male and female aged mice [18,19,20].

In addition to inactivity, vitamin D insufficiency (serum 25-OH Vitamin D < 30 ng/mL) is a prevalent problem affecting an estimated 50–70% of the world’s population. Vitamin D is essential for multiple cellular processes, including cell proliferation and differentiation [21], apoptosis [22], and angiogenesis [23]. It also plays a key role in regulating calcium homeostasis, which is crucial for muscle development and function [24]. The myriad biological actions of vitamin D are primarily mediated through interaction with the vitamin D receptor (VDR), a member of the superfamily of nuclear receptors that respond to steroid hormones and function as ligand-activated transcription factors, influencing gene expression and physiological processes [25]. We have previously demonstrated that vitamin D insufficiency increases frailty in mice as they age and reduces physical performance [26]. Furthermore, studies have shown that inactivating VDR contributes to muscle weakness [27,28,29].

However, the presence of VDR in muscle tissue has sparked considerable debate among researchers [30,31]. This controversy stems from earlier studies showing that many antibodies used to detect VDR produced visible bands on Western blots, even when VDR protein was absent [32]. Shortly thereafter, compelling evidence of VDR’s presence in muscle was provided, utilizing well-validated antibodies specific to mouse [32,33] and human muscle tissues [34]. Interestingly, while in healthy uninjured muscle, the level of VDR expression is notably low, it has been demonstrated that a single bout of moderate intensity exercise increases VDR expression in the muscle tissues of rats and horses [35,36]. However, the potential for HIIE to upregulate VDR expression, particularly during shorter session times, has not been investigated. Given the critical role of VDR in muscle regeneration and immune modulation, investigating its temporal expression pattern following exercise is essential for understanding its potential contribution to muscle adaptation and repair. To examine the temporal effects of exercise on VDR expression in skeletal muscle, we evaluated systemic and muscle-specific impacts 1, 4, and 24 h following a single bout of HIIE in aged male mice.

## 2. Materials and Methods

### 2.1. Animals

All studies and experimental protocols were approved by and in compliance with the guidelines of the Veterans Affairs Kansas City Medical Center Animal Care and Use Committee (protocol number: 1706183, approved 21 October 2022). At all times, mice were provided ad libitum access to food and water and were individually housed. Eighteen-month-old C57BL/6J mice were acquired (N = 33) from the Jackson Laboratory Aging Mouse Colony and, at 24 months of age, were randomly assigned to either sedentary (SED; *n* = 9) or high-intensity interval exercise groups (*n* = 8 for each post-exercise time-point). HIIE was performed on an inclined treadmill (25°) and started with a 3-min warm-up period at 5 m/min. This was followed by 20 s of high-intensity speed, then 20 s of active recovery at 5 m/min. High-intensity speeds started at 8 m/min, then 10 m/min, and then increased by 1 m/min thereafter. Mice were encouraged to continue running with a soft brush until they became exhausted (~10 min, defined as the mouse failing to respond to three consecutive motivational attempts). We have used and validated this endurance protocol in both older male and female mice [18,19,20]. Body weight was measured prior to exercise. 1, 4, and 24 h after HIIE, and for sedentary mice, blood and tissues were collected.

### 2.2. Serum Inflammatory Cytokines

Blood samples were centrifuged at 2000× *g* for 10 min using a Beckman Allegra 6R refrigerated benchtop centrifuge, allowing for the collection and aliquoting of serum. Cytokines were measured using a Bio-Techne ELLA multiplex ELISA system (Minneapolis, MN, USA) for IL-1β, IL-2, IL-5, IL-6, interferon gamma (IFNγ), transforming growth factor beta (TGF-β), and tumor necrosis factor alpha (TNF-α) from Bio-Techne (Kit ID: 210038 and 209219). Analysis of plates was performed using the Bio-Techne ELLA system software (Simple Plex Runner version 5.0.0.4).

### 2.3. Quantitative RT-PCR

Total RNA was isolated from the gastrocnemius muscle of sedentary, 1-, 4-, and 24-h groups (Table 1) using RNeasy Mini Kits (catalog #74104, Qiagen, Hilden, Germany) and subsequently, 500 ng of RNA was used for cDNA synthesis using SuperScript VILO cDNA Synthesis Kit (#11754050, Thermo Fisher Scientific, Waltham, MA, USA) as recommended by the supplier. The expression of the VDR gene was assessed using SBYR Gene Expression Assays (catalog #4453320). GAPDH was used as an endogenous control in the reaction. Each sample was run in triplicate. Delta (Δ) Ct values were used to determine relative expression changes (fold change, 2-ΔΔCT). Samples in RT-PCR analysis were compared using one-way ANOVA, and *p*-values < 0.05 were considered statistically significant.

### 2.4. Western Blotting

Gastrocnemius muscles in lysis buffer containing 10 mM Tris-HCL, 1% Triton X-100, 0.5% NP40, 150 mM NaCl, 10 mM Na orthophosphate, 10 mM Na pyrophosphate, 10 mM Na orthovanadate, 100 mM NaF, 1 mM EDTA, 1 mM EGTA, and 1× Protease Inhibitor Cocktail tablet (Roche, Cat No: 11836170001, Basel, Switzerland) (pH 7.45) were homogenized in a bullet blender 24 Gold at 40 C for 9 min. After centrifugation, supernatants were transferred to fresh tubes, and protein concentrations were determined with the Quick Start Bradford 1× Dye Reagent Kit (Bio-Rad #5000205, Hercules, CA, USA). Lysates were heated at 95 °C for 5 min. The proteins were then separated by 10% SDS-PAGE electrophoresis for 2.5 h at 120 V. Proteins were transferred onto polyvinylidene fluoride (PVDF) membranes for 80 min at 115 V, and the membranes were blocked with 5% skim milk powder in TBS plus 0.1% Tween 20 (TBST) for 1 h at room temperature. The membranes were then incubated overnight at 4 °C with primary antibody (Rabbit mAb, D2K6W, Cell Signaling Technology, Danvers, MA, USA) at a 1:1000 dilution in 2% skim milk powder in TBST. After three 5-min washes in TBST, membranes were incubated with a horseradish peroxidase (HRP)-conjugated anti-rabbit IgG secondary antibody (Vector Laboratories, PI-1000, Newark, CA, USA) at a 1:2000 dilution in 2% skim milk powder in TBST for 1 h at room temperature. After another set of three 5-min washes in TBST, bands were visualized using the SuperSignal West Femto Maximum Sensitivity Substrate (34096, Invitrogen, Waltham, MA, USA) and imaged with a ChemiDoc MP Image (Bio-Rad).

### 2.5. Tissue Processing and Immunofluorescent Assay

Mice were euthanized, and tibialis anterior (TA) muscles were collected, snap-frozen in liquid nitrogen-cooled isopentane for 6–10 s, and stored at −80 °C for at least 24 h to prepare samples for cryosectioning. Isopentane-treated TA muscles at −80 °C were then equilibrated in the cryostat chamber (−20 °C) for at least 30 min. Tissues were transversely cut at mid-belly and mounted in OCT with the cut surface facing upward. Sections (10 μm) were fixed in 3.7% formaldehyde with 1% Triton X-100 (Alfa Aesar, Haverhill, MA, USA) for 30 min at room temperature. Slides were washed 3 times for 3 min in PBS containing 0.1% Tween-20 (PBST). Sections were blocked for 1 h at room temperature in block buffer (10% FBS, 6% BSA, 0.1% Tx-100 in 1× PBS). Primary antibodies were then applied and incubated overnight at 4 °C. After washing three times for 5 min in PBST, secondary antibodies were added and incubated for 1 h at room temperature (Table 2). Sections were washed three times for 5 min each in PBST. Afterward, 2–3 drops of DAPI-containing mounting medium (Vectashield, Vector Laboratories, Newark, CA, USA) were added, then slides were imaged using a Keyence BZ-X series microscope. For the VDR marker, quantification of at least two sections per sample was analyzed and averaged to generate a single data point for the comparison between treatment groups. All quantification was performed by an investigator blinded to the identity of the samples.

### 2.6. Library Construction and Sequencing

Libraries were prepared using the TruSeq Stranded RNA Library Pre Kit following the manufacturer’s protocol. Per-cycle basecall (BCL) files generated by the Illumina NovaSeq 6000 (Illumina, San Diego, CA, USA) were converted to per-read FASTQ files using bcl2fastq version 2.20.0.422 using default parameters. The quality of the sequencing was reviewed using FastQC version 0.11.9. Detection of potential contamination was performed using FastQ Screen version 0.14.1. FastQC and FastQ Screen quality reports were summarized using MultiQC version 1.9. Samples received between 24.3 million and 9.1 million paired-end reads. Genomic alignments were performed using HISAT2 version 2.2.0 using default parameters. Ensembl reference GRCm38 was used for the reference genome and gene annotation set. Sequence alignments were compressed and sorted into binary alignment map (BAM) files using samtools version 1.16.1. Total alignment rates were between 83.3% and 67.1%. Counting of mapped reads for genomic features was performed using Subread feature Counts version 2.0.4 using the parameters -T 12 -p –count ReadPairs -s 2 -g gene_name -t exon -Q 60 -B -C, the annotation file specified with –a was the Ensemble GRCm38 annotation provided by Illumina’s iGenomes. Mapped reads assigned to gene features ranged from 11.2 million to 3.7 million. Alignment statistics and feature assignment statistics were summarized using MultiQC. Read count normalization using the regularized log transformation and principal component analysis were performed in R 4.2.2 using the Bioconductor package DESeq2 version 1.38.3. Principal component analysis was used to decompose the variation in the samples’ expression profiles into a set of uncorrelated variables of lower dimensions called PCs, with the first PC accounting for the most significant part of the total variation in the expression profiles and the subsequent PCs explaining less in decreasing order.

### 2.7. Data Analysis

All quantification and data analyses were performed by an investigator blinded to the identity of the samples. All statistical analyses were performed using GraphPad Prism (Version 10.4.1). Data were compared by one-way ANOVA, where appropriate significance was considered at *p* < 0.05. Each figure legend contains the statistical test and significance values. Data are reported as the mean ± SD.

## 3. Results

### 3.1. A Single Bout of HIIE Potentiates the Systemic Inflammatory Cytokine Profile in a Time-Dependent Manner in Aged Mice

The experimental design is illustrated in Figure 1A, and the body weight was measured before sacrifice (Figure 1B). To understand systemic impacts of HIIE, we first assessed a broad spectrum of pro-inflammatory and anti-inflammatory cytokine profiles, including IL 1β, IL 2, IL-5, IL-6, interferon-gamma (IFN-γ), TGF-β, and TNF-α. We found that serum IL 1β, TGF β, and TNF α levels in the 4-h post-HIIE group were significantly increased compared to the sedentary group, but this increase returned to baseline 24 h post-HIIE (Figure 1C). Moreover, significant increases in the serum levels of IL-6 were observed in the 1- and 4-h groups compared to the sedentary group, which also returned to baseline 24 h post-HIIE.

### 3.2. HIIE Significantly Temporally Alters Gastrocnemius Muscle Gene Expression

We next examined changes in muscle mRNA gene expression in response to HIIE. Principal component analysis (PCA) was performed to assess variability in gene expression across groups, displaying relationships among samples based on their expression profiles. Using PCA, we identified clear clusters that separated sedentary groups from those measured at 1 and 4 h after HIIE, which appear to resolve by 24 h (Figure 2A). Next, heat map analysis illustrates the 100 most significantly up- or down-regulated genes in sedentary and post-acute exercise mice, showing log2-fold changes in gene expression values across each sample (Figure 2B). The red-to-blue color scale represents the intensity of fold changes for each gene, with red indicating an increase in expression and blue indicating a decrease. Each row represents an individual gene, while each column shows the groupings of sedentary and HIIE. Interestingly, fourteen genes associated with myoblast differentiation were differentially expressed (Lrrc8a, Nmrk2, Sox9, Btg1, Sra1, Ccl9, Myf6, Boc, Flt3l, Cdon, Mapk14, Eid2b, Actl6a, and Smarcd1) 4 h after HIIE when compared to muscle from sedentary mice (Figure 2C & Appendix A). In contrast, there were no significant changes in the expression of genes associated with myoblast differentiation at 1 and 24 h compared with muscles from sedentary mice (Appendix A).

### 3.3. Increased VDR Expression Is Observed in Skeletal Muscle Tissue 4 h Following HIIE

Real-time reverse transcription–polymerase chain reaction (RT-PCR) analysis confirmed a significant increase in VDR expression in skeletal muscle tissue at 4 h post-HIIE (3.4 ± 1.1-fold) compared to the sedentary group (set as 1.0-fold), indicating a transcriptional response to acute exercise (Figure 3A). Western blot analysis also showed a substantial rise in VDR protein levels at 4 h post-HIIE; however, the levels of VDR protein remained elevated at 24 h (Figure 3B,C). We note, however, that high Ct values were observed during qPCR, and Western blotting required long exposure times, suggesting VDR content in muscle was on the threshold of detection. Immunohistochemical analysis was then used to identify where VDR is located within skeletal muscle fibers. In sedentary mice (Figure 3D), VDR expression (red) was mainly observed at the periphery of muscle fibers, shown with laminin. However, following HIIE, a prominent increase in the overall presence of VDR and nuclear localization (DAPI, blue) was apparent at 4 h post-exercise, and remained elevated at 24 h (Figure 3E).

### 3.4. VDR Expression Co-Localizes with Satellite Cells and F4/80-Expressing Macrophages Within Skeletal Muscle Tissue

Immunohistochemical analysis revealed that following HIIE, VDR expression (red) was strongly co-localized with PAX7-expressing satellite cells (yellow) and myonuclei (DAPI, blue). In sedentary mice (Figure 4A), VDR expression in satellite cells was minimal; however, a time-dependent increase was apparent following exercise, particularly at 4 and 24 h. Further analysis demonstrated that VDR expression was also identified in F4/80-expressing macrophages (yellow, Figure 4B). Specifically, muscles from mice 24 h post HIIE exhibited an apparent increase in the number of VDR co-localizations with F4/80 expression macrophages compared to the sedentary group.

## 4. Discussion

Low serum 25-OH Vitamin D levels have been associated with poor physical function, suggesting an important role of vitamin D in skeletal muscle [40,41]; however, the presence of VDR in this tissue has long been debated [30,31]. In a review, Pike [30] refers to Girgis et al. [33], who first reported that VDR is highly expressed in mouse neonate muscle tissue, and later exhibits lower expression in mice 3 months of age. This study also identified that VDR is highly expressed in satellite cells, but expression declines as cells differentiate towards myoblasts and myotubes [33]. More recently, it was demonstrated first in horses and then in rats that a single bout of moderate-intensity exercise increased VDR expression in muscle tissue [35,36]. Our study now extends this finding to demonstrate that a single bout of HIIE also increases VDR expression in mouse muscle tissue. However, a critical finding from this study is that the apparent increase in VDR expression may not be due to the skeletal muscle fibers but, rather, interstitial cells that are important for remodeling following exercise, including muscle satellite cells and macrophages. It remains unknown whether exercise specifically increases VDR expression within these cell types or if the identified tissue increases are simply due to their recruitment to the muscle tissue. We also note that our detection of VDR in muscle tissue was at the limits of sensitivity for both qPCR and Western, supporting the notion that VDR is not widely expressed in the fibers themselves.

Furthermore, our findings also include a delayed response in VDR upregulation, with increases being observable 4 h following exercise and remaining 24 h after exercise, in contrast to differences in mRNA after just 1 h, peaking at 4 h, and returning to baseline at 24 h. These findings support the notion that overall mRNA signaling responses to exercise may not be dependent on VDR signaling, which would be consistent with VDR residing not in muscle fibers, but in interstitial cells. The importance of VDR signaling in these cells might be highlighted by a study by Bass et al., which used a transient knockdown of the vitamin D receptor using intermuscular injection of short-hairpin RNA constructs [42]. This strategy, which potentially could also impact interstitial cells like satellite cells and macrophages, resulted in autophagic upregulation and myofiber atrophy. However, despite the lack of VDR in muscle fibers suggested by our findings, several studies support the potential for muscle fibers to be responsive to VDR. This includes a study that overexpressed VDR via electrotransfer of plasmids in the muscle tissue, stimulated hypertrophy in rats [43], suggesting muscle fibers have the propensity to respond to VDR signaling. Further, a muscle-specific knock-out of VDR, which would theoretically target myofibers specifically, resulted in reduced grip strength and lower muscle mass [43]. Although this latter study suggests that, despite low VDR expression, there is a significant biologic impact in myofibers, the contributions during embryonic development may not be ruled out.

Our findings demonstrate the dynamic regulation of VDR expression in skeletal muscle satellite cells and macrophages following HIIE. Immunohistochemical analysis revealed a significant upregulation of VDR^+^ PAX7^+^ satellite cells at 4 h post-exercise, which returned to baseline by 24 h. This transient increase suggests that VDR may play a role in the early activation of muscle progenitor cells, potentially influencing muscle repair and adaptation to exercise-induced stress. The observed increase in VDR^+^ satellite cells aligns with previous studies indicating that VDR signaling is involved in myogenesis and muscle regeneration [33]. Given that satellite cells are essential for skeletal muscle repair, the upregulation of VDR at 4 h post-HIIE suggests that vitamin D signaling may be an essential modulator of early-stage muscle recovery. The return to baseline levels by 24 h indicates this activation is transient, likely corresponding to the early phases of satellite cell proliferation and commitment to differentiation [44,45]. Satellite cells are identified as skeletal muscle stem cells that reside between the basal lamina and sarcolemma, and they play a significant role in skeletal muscle growth and regeneration [46,47]. Generally, satellite cells exist in a quiescent state in homeostatic conditions and are activated by stimuli such as injury or exercise, for skeletal muscle regeneration or hypertrophy. Activated satellite cells will become myoblasts, which proliferate and differentiate further to form new muscle fibers or fuse with preexisting muscle fibers [46,48]. Hence, studying VDR expression in satellite cells could reveal the potential role of VDR in skeletal muscle mass and regeneration following acute HIIE.

In contrast, VDR expression in F4/80-expressing macrophages exhibited a distinct temporal pattern, remaining elevated 24 h post-HIIE. This suggests a potential role for VDR in the immune response and the resolution of inflammation following acute exercise. Macrophages are known to orchestrate muscle regeneration by secreting cytokines and growth factors that regulate the activity of satellite cells [49]. The prolonged expression of the M2 macrophage phenotype may indicate its involvement in modulating the immune environment of the regenerating muscle, potentially facilitating the shift from a pro-inflammatory (M1) to an anti-inflammatory (M2) phenotype, which is crucial for efficient muscle repair [50]. These findings further prove that VDR is dynamically regulated in response to acute exercise, playing a dual role in early satellite cell activation and the later immune-mediated muscle repair process. Future studies should investigate whether vitamin D supplementation can enhance these effects, potentially improving recovery and adaptation to acute HIIE and HIIE training programs.

It is well established that low-grade inflammation is associated with aging, as proinflammatory cytokines are increased [51]. This study also examined the relationship between HIIE and the differential modulation of pro- and anti-inflammatory cytokines in aging. We found that TNF-α, IL-1β, and IL-6 are significantly increased at 4 h following HIIE, with IL-6 also being increased after just 1 h. It has been shown that IL-6 is associated with muscle loss, decreased bone mass, impaired performance, and balance [52]. IL-6 is the most widely studied cytokine, as it plays a crucial role in various immunological, inflammatory, and metabolic functions [53,54].

Furthermore, it has been reported that both TNF-α and IL-1β strongly promote IL-6 secretion. In contrast, the increase in IL-6 secretion also serves as a feedback mechanism, degrading receptors and releasing TNF-α and IL-1β, resulting in anti-inflammatory effects [55], which is consistent with a sustained increase in IL-6 following HIIE, potentially promoting an anti-inflammatory effect. TGF-β has also been reported as an exercise-induced cytokine that improves glucose tolerance and insulin sensitivity, increases fatty acid uptake and oxidation, and stimulates glucose uptake in skeletal muscle, the heart, and BAT [56]. This is also consistent with our data on the significant increase in circulatory TGF-β levels 4 h following HIIE and may well warrant further investigation into the roles of anti-inflammatory cytokines in repeated exposure to HIIE.

Additionally, mRNA sequencing analysis of gastrocnemius muscle tissues following HIIE provides valuable insights into the temporal transcriptional responses. The PCA revealed distinct clustering of gene expression profiles at 1- and 4-h post-exercise, with a return to baseline levels by 24 h. This suggests that the most pronounced transcriptional changes occur within the early hours following exercise, supporting the idea that acute bouts of HIIE induce rapid but transient molecular adaptations in skeletal muscle. Heat map analysis of the 100 most significantly up- or down-regulated genes further highlights the dynamic nature of the transcriptomic response. Notably, we identified 14 differentially expressed genes at the 4-h post-exercise time point, which are implicated in myoblast differentiation. These include key regulators of muscle regeneration and remodeling, such as Myf6, Sox9, and Actl6a, which influence myogenic lineage commitment and cellular proliferation. The upregulation of Lrrc8a and Nmrk2 suggests an enhanced cellular stress response, potentially linked to osmotic regulation and NAD^+^ metabolism, which play roles in muscle adaptation and recovery.

Interestingly, genes such as Boc, Flt3l, and Cdon were also differentially expressed in signaling pathways regulating myogenesis and extracellular matrix interactions. This indicates that acute exercise may transiently activate muscle remodeling and repair pathways. Moreover, Mapk14, a component of the p38 MAPK signaling pathway, was differentially expressed, consistent with previous findings linking this pathway to muscle adaptation and the stress response following exercise. The observed return to baseline gene expression at 24 h post-exercise suggests that the transcriptional changes induced by HIIE are short-lived, raising the question of whether repeated exercise bouts result in sustained molecular adaptations. These findings align with prior research indicating that transient transcriptional activation following acute exercise contributes to the development of cumulative physiological adaptations over time [57,58].

The work herein raises new possibilities for VDR signaling in muscle tissue in response to exercise, but some limitations should be recognized as well as possible new directions for this research. First, this study involves aged mice, and we note these data does not support or refute the safety of HIIE for older adults. However, short session HIIE has been successfully implemented in older adults [59,60]. Furthermore, two reviews support the safety of HIIE, including with reduction in the long-term risk for falls [61,62], although there was some concern for transient risk in the initial sessions [62]. Additionally, this investigation is limited to muscle groups with higher composition of fast twitch fibers, which may not reflect the responses of predominately slower twitch muscle found in humans. This study also lays the foundation for further investigation of the role of VDR in ancillary cells, with the use of VDR KO mice and deeper characterizations of macrophage M1/M2 or mixed phenotypes and satellite cells. Future work would also be needed to investigate if HIIE is effective in inducing VDR expression in younger animals. We do note that moderate exercise successfully induced VDR upregulation in young adult rats and horses [35,36].

## 5. Conclusions

Overall, our study provides a comprehensive characterization of the early molecular responses to a single bout of HIIE, emphasizing key regulators of myogenesis and skeletal muscle adaptation. Our findings suggest VDR upregulation in skeletal muscle follows overall mRNA and cytokine profile changes, and the presence of VDR is more likely attributed to the proliferation of satellite cells and recruitment of macrophages in the muscle tissue—which may indicate a VDR role in remodeling following exercise. Future research should investigate the functional implications of these transient transcriptional changes and examine whether repeated exercise bouts lead to cumulative shifts in gene expression that promote long-term muscle remodeling and performance enhancements.

## Figures and Tables

**Figure 1 nutrients-17-03733-f001:**
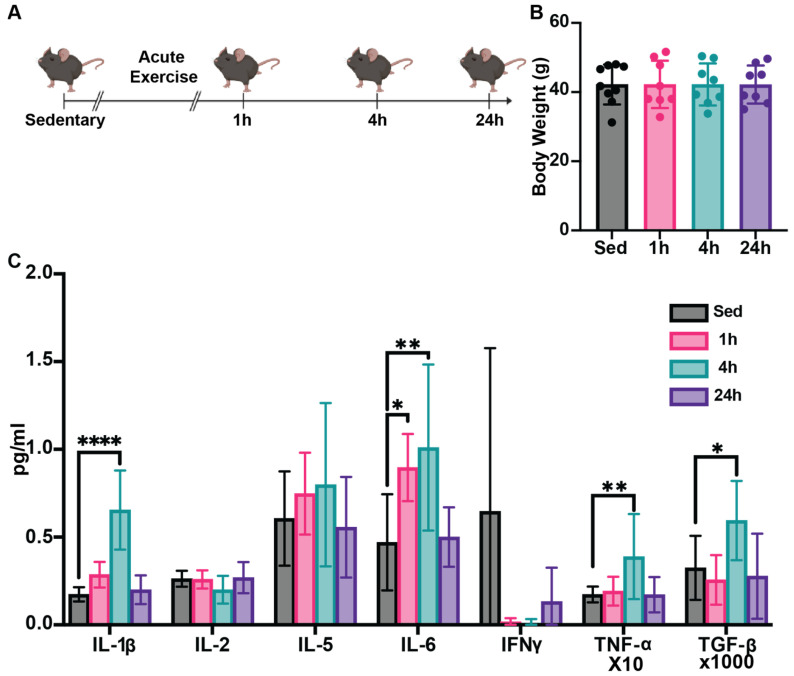
HIIE influences cytokine profiles in aged mice. (**A**) Experimental design for this experiment (*n* = 9 for the sedentary group and *n* = 8 for the each of the 3 exercise groups). Male C57Bl6/J mice (24 months of age) were provided a single 10-min session of HIIE, or left sedentary, and then tissues were harvested 1-, 4-, and 24-h later. (**B**) The body weight of the mice in each group was measured prior to sacrifice. (**C**) Serum was then assessed to generate an inflammatory cytokine profile using the Bio-Techne ELLA multiplex ELISA system. Mean ± SD; * *p* < 0.05, ** *p* < 0.01, **** *p* < 0.0001.

**Figure 2 nutrients-17-03733-f002:**
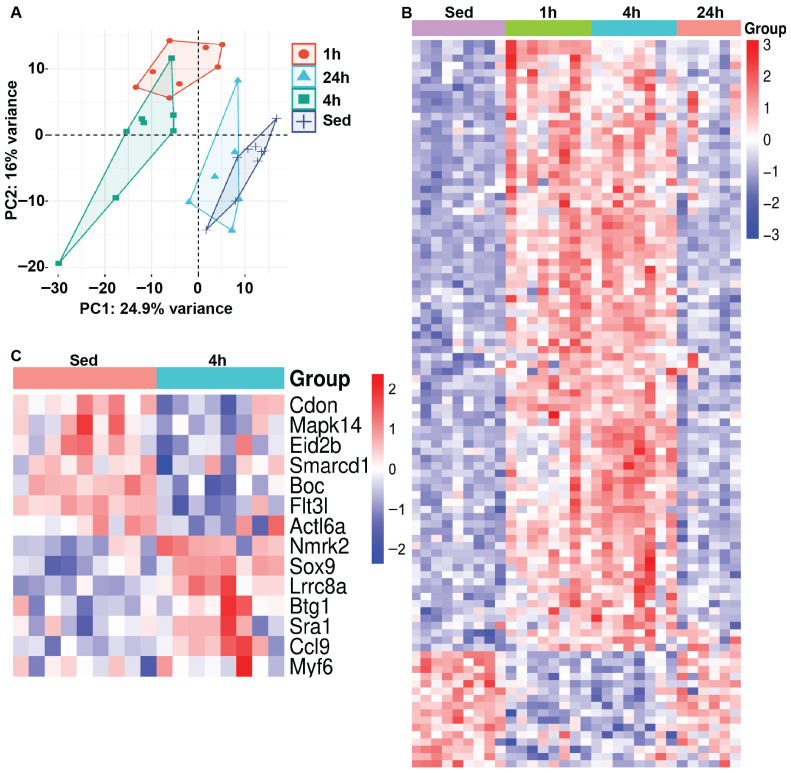
Gene expression profiles are directly influenced by acute exercise. RNA sequencing was conducted on gastrocnemius muscle tissues following a single bout of HIIE and used for (**A**) principal component analysis (PCA) (*n* = 9 for the sedentary group and *n* = 8 for each of the 3 exercise groups). The samples from the 1- and 4-h post-acute exercise mRNA profiles (rlog-transformed, 50 most variable) are segregated, with samples from the 24-h post-acute returning to baseline/sedentary control levels. (**B**) Heat map analysis illustrates the 100 most variably expressed genes in sedentary and post-acute exercise mice, showing log2-fold changes in gene expression values across each sample. (**C**) Heat map analysis illustrates the variably expressed genes associated with myoblast differentiation in sedentary and 4-h post-acute exercise mice, demonstrating log2-fold changes in gene expression values across each sample.

**Figure 3 nutrients-17-03733-f003:**
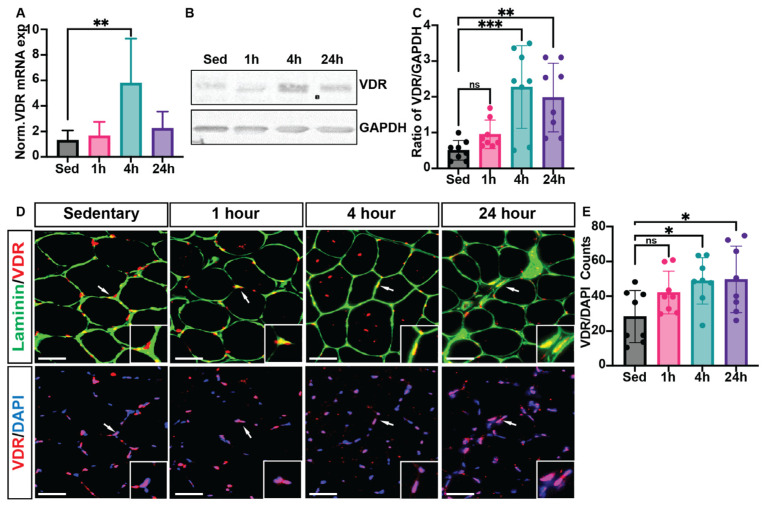
VDR expression is enriched in skeletal muscle tissue following 4 and 24 h of acute HIIE. Following a single bout of HIIE, RNA was isolated from the gastrocnemius muscle to evaluate (**A**) VDR mRNA and protein extracts, and (**B**) VDR expression using Western blots. (**C**) Quantification of VDR protein expression was further analyzed using ImageJ. Immunofluorescent analysis (**D**) was then used to determine the localization of VDR expression in the mouse tibialis anterior muscle. In all images, laminin (green) denotes the fiber border, myonuclei (stained with DAPI, blue), and VDR expression (red) were analyzed. (**E**) VDR content was then analyzed as the number of VDR/DAPI co-localizations per section (with 2 sections averaged per animal). Data are expressed as mean ± SD, with *n* = 8 mice per group. Mean ± SD; * *p* < 0.05, ** *p* < 0.01, and *** *p* < 0.001, respectively. Scale: bar = 100 μm. ns—not significant. White arrows denote area magnified in corner box.

**Figure 4 nutrients-17-03733-f004:**
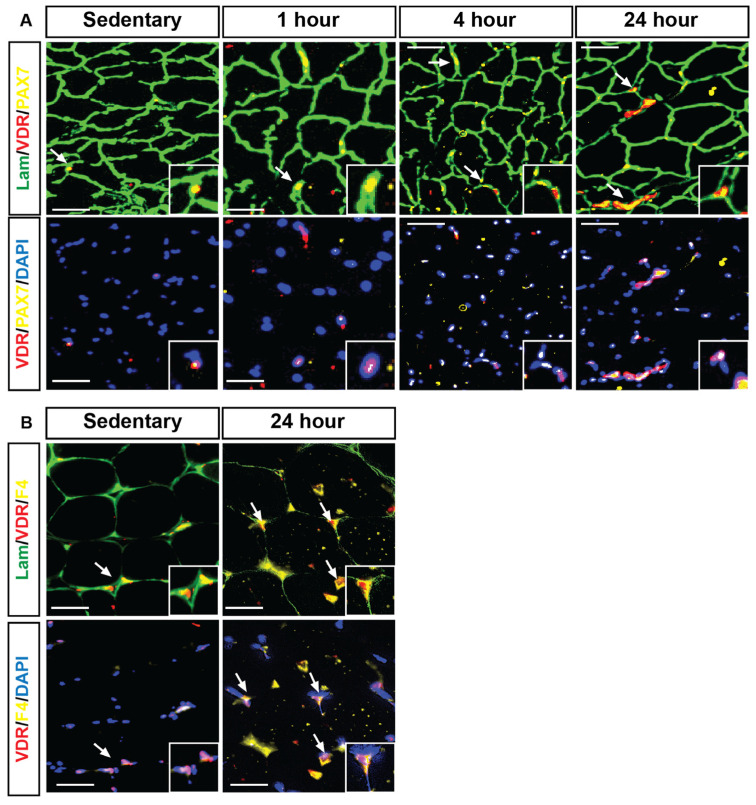
VDR expression was identified within satellite cells and macrophages, both showing an apparent increase 24 h following HIIE. Immunofluorescent images illustrate that laminin (green) marks the fiber boundaries, while centrally located myonuclei (DAPI, blue) show strong positive VDR expression (red puncta) co-localizing with PAX7-expressing satellite cells (yellow) and myonuclei. (**A**) (sedentary, 1 h, 4 h, and 24 h) depicts the co-localized expression of VDR and PAX7 at the fiber borders. In contrast, (**B**) (sedentary, 1 h, 4 h, and 24 h) shows the co-localization of VDR and PAX7 with the myonuclei. Next, VDR expression (red) co-localization with F4-expressing macrophages (yellow) and myonuclei (DAPI, blue) was investigated within the skeletal muscle fibers (laminin, green), assessed by immunofluorescent analysis in the sedentary and 24-h post groups after HIIE. VDR expression is enriched in skeletal muscle tissue following 4 and 24 h of acute HIIE. Scale: bar = 100 μm. White arrows denote co-localized VDR with either PAX7 (**A**) and F4 (**B**).

**Table 1 nutrients-17-03733-t001:** List of primers.

Gene	Primer Sequence (5′–3′)
VDR	F-TGTGACCCTGGACCTCTCTC
	R-AGAGGTGAGGTCCCTGAAGC
Gpi	F-ATCGCCTCCAAGACCTTCAC
	R-AACTGCAGATGGATCCTTGG
Cdon	F-TCCTTGTGAGTCGTCCTTCC
	R-ACTCACCACACACTCCAAGG
Smarcd1	F-ACCTCAACAGATCCAGCAGG
	R-GACCAGTTCCCGAATCCTTTG
Flt3l	F-GTGAAGTTTAGAGAGTTGACTGACC
	R-ATCTCGGTGTTGACGTCCTC
Nmrk2	F-CTTCTTCAAGCCCCAGGACC
	R-AGGAGGAGTACGTGGGTGTC
Lrrc8a	F-TGCAGAACCTCCAGAACCTG
	R-TTGTTGCCCAGGTGTAGAGC
CCl9	F-CCGGGCATCATCTTTATCAGC
	R-TGTAGGTCCGTGGTTGTGAG

**Table 2 nutrients-17-03733-t002:** IF antibodies.

Antibody	Cat#/Vendor	Ratio
VDR (Vitamin D Receptor) [37]	MA1-710/Thermo Fisher	1:200
Pax 7 [38]	PA1-117/Thermo Fisher	1:50
Laminin [38]	Ab11575/Abcam (Cambridge, UK)	1:600
F4/80 (macrophage) [39]	MA5-47762/Thermo Fisher	1:100

## Data Availability

The original contributions presented in this study are included in the article/Appendix A. Further inquiries can be directed to the corresponding author.

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
