# Peer review of "High-Intensity Interval Exercise Drives Vitamin D Receptor Expression in Skeletal Muscle via Recruitment of Non-Parenchymal Cells, Not Upregulation in Muscle Fibers"

_nutrients, 2025, doi:10.3390/nu17233733_

Round 1

Reviewer 1 Report

Comments and Suggestions for Authors

In this manuscript, the authors have demonstrated the increase in VDR at mRNA and protein level in the skeletal muscle tissue upon high intensity interval exercise. The authors used the old sedentary mice. The exercise was of 10 minutes and the authors observed the increased upregulation of VDR at 4h (mRNA and protein) and 24 h (protein). The authors perform the high throughput sequencing analysis to determine the changes in gene expression suggesting that the exercise is leading to changes in myoblast gene expression. The study is interesting and the idea is quite good. The selection of aged mice and the linking of VDR is an important connection. It will have relevance for the field of aging and muscle degeneration with it. The manuscript is well written and the authors have discussed the limitation of the study. However, there are several demerits in the study which needs to be addressed before its final acceptance for publication. My comments are provided below

  1. The title and abstract are well written. In the abstract the authors have mentioned that ‘greater VDR expression following exercise may be attributed to ancillary 28 cells that respond to muscle remodeling following acute exercise’. However, there is no such data in the manuscript. In Fig 3 and Fig 4, the analysis was on the types of cells that express VDR. But the comparison between different cell types is missing. I would suggest the authors to reconsider this statement.
  2. The introduction section is well written and the references are good. However, the link between VDR and high intensity interval exercise (HIIE) is missing. The authors described HIIE and its potential for replacing normal exercise and then describe VDR. However, the logic for connection is missing. I would suggest the authors to elaborate on it.

Method

The method section is well written and the authors have provided sufficient details for the elaboration.

  1. The animal housing condition is well written but the authors did not provide the animal ethics number and approval date.
  2. Please provide the catalog number of the ELISA kit.
  3. Please describe the RNA isolation procedure in details and the quality control for it. Please mention the amount of RNA used for cDNA preparation. Please provide the primer information.
  4. Please provide the catalog number of Protease Inhibitor Cocktail tablet.

Result

The authors localized the VDR in the skeletal muscle tissue upon exercise. The use of two different methods to determine the changes is making the study reliable. One of the important drawbacks of the study is the use of fast acting fibres and lack of use of VDR KO mice. The use of aged mice is somehow limiting its generalization to other population. Further the study is lacking the data relevant to human limiting its clinical relevance. The functional relevance of the findings is missing which the authors have acknowledged. The detailed characterization of the cells present in the muscle tissue is missing. For example, the authors have used only one marker for each cell types. The macrophages are also not characterized in details. The study did not provide what percentage of macrophage are M1 or M2 or mixed phenotypes. The characterization of satellite cells also needs more detail study.

Figure 1

The authors determine the cytokines profile from the serum upon exercise. I would suggest the authors to perform gene expression study of the same genes using muscle tissue. This will support the data for macrophage recruitment to the skeletal muscle.

Figure 2

I would suggest the authors to validate the RNA sequencing data by RtqPCR. Since the authors have performed the RNA sequencing using the tissue, I would suggest the characterization of the cell types changes using machine learning based model. This will support the data for Fig 4 whether there is increase macrophage or not.

Please also mention the biological and technical replicates in the figure legend.

Figure 3

For Fig 3B, I would suggest the authors to normalize the expression with total protein.

Please mention the molecular size markers in the Fig 3B.

Please provide the scale bar in the Fig 3D.

Figure 4

Please mention the scale bar in all the figures. Please mention the number of fields per mouse were imaged.

Please provide a quantitative data for the figures.

Discussion

The discussion is nicely written. I would suggest the authors to mention those things as the limitation of the study which they couldnot perform due to any limitation.

I would suggest the authors to discuss about the potential role of this study on life expectancy and muscle function (strength and NMJ function) and its relevance in Duchenne muscular dystrophy considering the VDR expression.

The benefits of HIIE on young mice needs to be discussed.

Author Response

Reviewer 1

In the abstract the authors have mentioned that ‘greater VDR expression following exercise may be attributed to ancillary cells that respond to muscle remodeling following acute exercise’. However, there is no such data in the manuscript. In Fig 3 and Fig 4, the analysis was on the types of cells that express VDR. But the comparison between different cell types is missing. I would suggest the authors to reconsider this statement.

We have restructured the sentence as follows: As VDR was localized to the periphery of muscle fibers, we investigated and found that VDR co-localizes with PAX7 (a marker for satellite cells) and F4/80-expressing macrophages. This suggests that increases in VDR expression following exercise may be attributed to ancillary cell response during muscle remodeling.  We believe this statement is supported by our observation that VDR co-localized with proliferating satellite cells and recruited macrophages as demonstrated in Figure 4, while not finding increases in VDR content within the muscle fibers themselves.

The link between VDR and high intensity interval exercise (HIIE) is missing. The authors described HIIE and its potential for replacing normal exercise and then describe VDR. However, the logic for connection is missing. I would suggest the authors to elaborate on it.

      We adjusted transitions of the first and second paragraphs to aid readability. Additionally, we structured the final paragraph to clarify the rationale for investigating HIIE and VitD signaling as: Interestingly, while in healthy uninjured muscle, the level of VDR expression is notably low, it has been demonstrated that a single bout of moderate intensity exercise increases VDR expression in the muscle tissue of rats and horses (37, 38). However, the potential for HIIE to upregulate VDR expression in muscle, particularly during shorter session times, has not been investigated.

The animal housing condition is well written but the authors did not provide the animal ethics number and approval date.

We have now added the following: (protocol number: 1706183, approved October 21st, 2022).

Please provide the catalog number of the ELISA kit. Please describe the RNA isolation procedure in details and the quality control for it. Please mention the amount of RNA used for cDNA preparation. Please provide the primer information. Please provide the catalog number of Protease Inhibitor Cocktail tablet.

These corrections have been made

One of the important drawbacks of the study is the use of fast acting fibres and lack of use of VDR KO mice. The use of aged mice is somehow limiting its generalization to other population. Further the study is lacking the data relevant to human limiting its clinical relevance. The functional relevance of the findings is missing which the authors have acknowledged. The detailed characterization of the cells present in the muscle tissue is missing. For example, the authors have used only one marker for each cell types. The macrophages are also not characterized in details. The study did not provide what percentage of macrophage are M1 or M2 or mixed phenotypes. The characterization of satellite cells also needs more detail study. Since the authors have performed the RNA sequencing using the tissue, I would suggest the characterization of the cell types changes using machine learning based model. This will support the data for Fig 4 whether there is increase macrophage or not. The benefits of HIIE on young mice needs to be discussed.

We thank the reviewer for the identified limitations and potential new directions for our study. Indeed, our focus on faster twitch gastrocnemius and tibialis anterior muscles raises questions as to the behavior of VDR in slower twitch muscles like the soleus during exercise. Further, additional characterization of satellite cells and M1/M2 macrophages might provide more insight into specific roles VDR has within these cell types. This latter aspect might be further elucidated by investigating acute exercise in a VDR KO model. The scope of our paper however was to investigate the phenomenon of VDR increase in muscle tissue in response to HIIE, and we look forward to future possibilities that extend from this work. Furthermore, we anticipate these biologic processes to be found in younger organisms as well as in humans – for which future work will be needed to confirm. Finally, the use of machine learning is an interesting one, and datasets will be made available to investigators who wish to collaborate to investigate this possibility. We appreciate the points raised here and have added a new limitations and future directions paragraph to share these ideas.

The authors determine the cytokines profile from the serum upon exercise. I would suggest the authors to perform gene expression study of the same genes using muscle tissue. This will support the data for macrophage recruitment to the skeletal muscle.

We performed qPCR analyses for several cytokines featured in Figure 1C, however our findings were at the threshold of detection, which is consistent with our genome wide mRNA analysis. Deeper investigation may be necessary that involves analysis of protein level expression, however, this is limited by resource/sample restraints at this time.

For Fig 3B, I would suggest the authors to normalize the expression with total protein.

We appreciate the reviewer’s suggestion and while we agree that total protein normalization can offer advantages in certain context, due to resource/sample restraints we are unable to implement this approach for the current study. Normalization using GAPDH is a widely accepted and validated method for normalization in muscle tissue. Although there are limitations, particularly in the context of older versus younger animals or fiber-type dependencies. We used similarly aged animals and generated lysates from a similar region (mid-belly) of the gastrocnemius for all animals.

I would suggest the authors to validate the RNA sequencing data by RtPCR.

Validations are now shown in Supplemental Figure 1.

Please also mention the biological and technical replicates in the figure legend. Please mention the molecular size markers in the Fig 3B. Please mention the scale bar in all the figures. Please mention the number of fields per mouse were imaged.

We have added these elements as suggested.

Please provide a quantitative data for the figures (Figure 4).

Quantitation would be difficult as we conducted only a representative sample from each group to ascertain the co-localization of VDR with indicators for satellite cells and macrophages. Additionally, the response of satellite cells and macrophages to exercise is well supported in the literature (PMIDs: 37923703, 23464362, 29470148, 25313863, 40879941), and quantitation would only be confirmatory.

The discussion is nicely written. I would suggest the authors to mention those things as the limitation of the study which they could not perform due to any limitations.

      We appreciate this suggestion and have added a new paragraph to the end of the discussion to elaborate on these aspects.

I would suggest the authors to discuss about the potential role of this study on life expectancy and muscle function (strength and NMJ function) and its relevance in Duchenne muscular dystrophy considering the VDR expression.

We appreciate the suggested areas for further discussion of the implications from this body of work. This work will be part of a special issue for which there will be a commentary.  That commentary will be authored by one of our team members (Dr. Seldeen), where he will have the opportunity raise these possibilities.

Reviewer 2 Report

Comments and Suggestions for Authors

The manuscript presents an interesting and well-conducted study investigating the effects of a single session of high-intensity interval exercise (HIIE) on vitamin D receptor (VDR) expression and systemic inflammatory response in aged mice. The methodology is solid, and the data are convincing, but several conceptual and editorial issues still need to be addressed before the paper can be accepted.

  1. Introduction section
    The Introduction currently starts with a long discussion on frailty and hospitalizations in elderly humans, which gives the impression that the study focuses on clinical subjects. In fact, all experiments were conducted on mice. This may easily confuse readers and set an incorrect context for the work.
    I recommend that the authors revise the Introduction so that it begins with the physiological background of exercise, aging, and VDR signaling in muscle. The part about human frailty could be shortened or moved to the end of the Introduction as a contextual note rather than the main rationale.
  2. Definition and interpretation of HIIT/HIIE
    In exercise physiology, HIIT (or HIIE) refers to repeated bouts of near-maximal intensity effort that lead to significant depletion of muscle glycogen and induce strong metabolic stress. This form of training is physiologically exhausting and may not be safe for older adults.
    Therefore, the suggestion that positive effects of HIIE in aged mice could be extrapolated directly to individuals over 65 years old may be misleading. I strongly recommend that the authors clarify in the Discussion that this is an animal model, and that such protocols should not be considered directly applicable or safe for elderly people. Otherwise, the conclusions might be interpreted as a kind of exercise recommendation, which would be inappropriate and potentially unsafe.
  3. Terminology inconsistency (HIIE vs. HIIT)
    The title uses the term HIIE (“high-intensity interval exercise”), but in the text both HIIE and HIIT appear interchangeably. This inconsistency may create confusion for readers. I suggest choosing one abbreviation (preferably HIIE, as in the title) and using it consistently throughout the manuscript.
  4. Missing reference to Table 1
    Table 1 lists antibodies used in immunofluorescent assays, but there is no clear reference to it in the text. Please add an in-text citation in the Methods section when describing the immunofluorescence protocol.
  5. Ethical considerations and ARRIVE checklist
    The ARRIVE checklist has been properly completed and included. The manuscript provides sufficient information on animal care, housing, and experimental procedures, and there are no apparent concerns regarding animal welfare or ethical compliance. The description follows the 3R principles and meets the expected reporting standards.

Moreover:

There are a few typographical and stylistic inconsistencies (e.g., inconsistent capitalization of “Vitamin D receptor”, mixed use of tenses, and small grammar errors). A careful proofreading would improve readability.

The Discussion might also briefly note the inherent limitations of translating findings from mice to humans, to make the interpretation more balanced.

Overall evaluation
This is a scientifically valuable paper with solid experimental design and relevant results. However, the manuscript requires revision to improve clarity, terminology consistency, and the framing of the Introduction and Discussion. After these issues are corrected, the paper could be suitable for publication.

Recommendation: Major Revisions Required.

Author Response

Reviewer 2
The Introduction currently starts with a long discussion on frailty and hospitalizations in elderly humans, which gives the impression that the study focuses on clinical subjects. In fact, all experiments were conducted on mice. This may easily confuse readers and set an incorrect context for the work. I recommend that the authors revise the Introduction so that it begins with the physiological background of exercise, aging, and VDR signaling in muscle. The part about human frailty could be shortened or moved to the end of the Introduction as a contextual note rather than the main rationale.
Thank you for this insight – we have now restructured the start of our introduction to frame the focus of this manuscript on exercise.
In exercise physiology, HIIT (or HIIE) refers to repeated bouts of near-maximal intensity effort that lead to significant depletion of muscle glycogen and induce strong metabolic stress. This form of training is physiologically exhausting and may not be safe for older adults. Therefore, the suggestion that positive effects of HIIE in aged mice could be extrapolated directly to individuals over 65 years old may be misleading. I strongly recommend that the authors clarify in the Discussion that this is an animal model, and that such protocols should not be considered directly applicable or safe for elderly people. Otherwise, the conclusions might be interpreted as a kind of exercise recommendation, which would be inappropriate and potentially unsafe. The Discussion might also briefly note the inherent limitations of translating findings from mice to humans, to make the interpretation more balanced.
We appreciate the concern and will note that there are multiple studies on the safety and efficacy of HIIT (HIIE) in older adults. In addition, we have successfully conducted a trial of HIIT in older Veterans that was both safe and beneficial and for which we are preparing a manuscript and have also received funding from the VA for a full clinical trial. Indeed, we have based our clinical trial on our published HIIT protocol for older mice. Therefore, we have added the following to the manuscript: … First, this study involves aged mice, and we note these data does not support or refute the safety of HIIE for older adults. However, short session HIIE has been successfully implemented in older adults [59, 60]. Furthermore, two reviews support the safety of HIIE, including with reduction in the long term risk for falls [61, 62], although there was some concern for transient risk in the initial sessions [62].
Terminology inconsistency (HIIE vs. HIIT): The title uses the term HIIE (“high-intensity interval exercise”), but in the text both HIIE and HIIT appear interchangeably. This inconsistency may create confusion for readers. I suggest choosing one abbreviation (preferably HIIE, as in the title) and using it consistently throughout the manuscript.
We have corrected the manuscript to use only HIIE to increase readability. Thank you.
Missing reference to Table 1: Table 1 lists antibodies used in immunofluorescent assays, but there is no clear reference to it in the text. Please add an in-text citation in the Methods section when describing the immunofluorescence protocol.
We have added these references.
There are a few typographical and stylistic inconsistencies (e.g., inconsistent capitalization of “Vitamin D receptor”, mixed use of tenses, and small grammar errors). A careful proofreading would improve readability.
Thank you for this note - we have improved the manuscript.

Round 2

Reviewer 1 Report

Comments and Suggestions for Authors

The authors have addressed all my concerns in the revised manuscript. I support the publication of the revise version of the manuscript.